# SimplexLoRA: Dynamic Rank Updating via Alternating Minimization with Simplex Projection

## Abstract

The success of language models in real-world applications depends on parameter-efficient fine-tuning methods such as Low-Rank Adaptation (LoRA), which inserts compact, trainable adapters into selected layers. We propose SimplexLoRA, a novel LoRA variant that adaptively adjusts adapter ranks during fine-tuning based on their learned importance. Unlike standard LoRA, which uses a fixed rank for all layers, SimplexLoRA optimizes adapter weights on a probabilistic simplex, dynamically allocating capacity where it is most needed without increasing the parameter budget. We also introduce efficient algorithms for rank expansion and compression using QR decomposition and truncated SVD. Experiments on the GLUE benchmark with DeBERTaV3 and Llama 3 models show that SimplexLoRA matches or surpasses standard LoRA and recent adaptive methods, while significantly reducing the number of trainable parameters. Our code is available at `https://anonymous.4open.science/r/SimplexLoRA-C142`.

## 1 Introduction

Large language models (LLMs) currently occupy a central position in modern machine learning and AI research (Raffel et al., 2020; Sanh et al., 2021), with fine-tuning techniques rapidly advancing in this field (Lester et al., 2021). One of the most widely used methods for fine-tuning pre-trained LLMs today is Low-Rank Adaptation (LoRA) (Hu et al., 2021). Introduced in 2021, LoRA significantly improved the efficiency and reduced the costs of model adaptation for downstream tasks, sparking a surge of research and further innovation in this direction.

The standard formulation of fine-tuning with LoRA involves modifying the weight matrices in several layers as follows:

$$\boldsymbol{W}' = \boldsymbol{W}_0 + \Delta\boldsymbol{W} = \boldsymbol{W} + \boldsymbol{AB},$$

where $\boldsymbol{W} \in \mathbb{R}^{m \times d}$ is a pre-trained weight matrix, $\boldsymbol{A} \in \mathbb{R}^{m \times r}, \boldsymbol{B} \in \mathbb{R}^{r \times d}$ are trainable parameters with $r \ll m, d$. This approach achieves no inference overhead and a little memory overhead during fine-tuning.

Due to its flexibility and efficiency, LoRA remains a widely sought-after method in both academia and industry. It enables efficient adaptation of models with a fixed architecture to highly specialized tasks by training only a small fraction of their parameters (Alva et al., 2025). This efficiency makes LoRA a go-to algorithm for fine-tuning, particularly suited for rapid task-solving. Moreover, its adjustable rank feature allows achieving metric values surpassing those of full fine-tuning, while the fast training speed facilitates exploration of large hyperparameter grids (Mehmood et al., 2025).

However, LoRA (Hu et al., 2021) employs a fixed-rank structure for all adapters, ignoring potential variations in layer-wise convergence behavior. Empirical evidence suggests that different layers converge to distinct local minima, exhibiting significant heterogeneity in their learned representations.

To address this limitation, we introduce an adaptive rank allocation strategy that dynamically adjusts the rank of each layer's adapter based on its estimated contribution to the fine-tuning process.

Our method formulates and solves an optimization problem for rank distribution across LoRAs to maximize overall model performance.

Based on these considerations, we propose a novel method for adaptive rank selection in LoRA, aimed at accelerating training and improving its effectiveness — SimplexLoRA (see Figure 1). Our method possesses following advantages:

• Ranks of LoRA adapters are adaptively trained during fine-tuning.

• Ranks are learned during a brief initial phase and then frozen, followed by standard LoRA fine-tuning, unlocking a wide range of existing LoRA techniques.

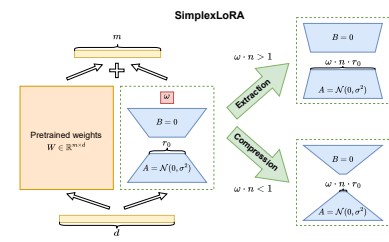

Figure 1: SimplexLoRA framework.

• We can enforce a parameter budget similar to standard LoRA by leveraging optimization on the simplex.

• SimplexLoRA enables further reduction of the total number of trainable parameters while maintaining competitive performance.

## 2 RELATED WORK

### 2.1 CLASSICAL LOW-RANK APPROACHES

While LoRA has popularized parameter-efficient fine-tuning, it is not the only method to explore low-rank or structured adaptations. Earlier approaches include LoHA (Hyeon-Woo et al., 2021), which utilizes element-wise (Hadamard) interactions between low-rank matrices to reduce the parameter count without sacrificing expressivity, and Krona (Edalati et al., 2022), which employs the Kronecker product in place of the standard matrix multiplication further enhancing parameter efficiency while maintaining model capacity. Since then, various methods have extended or refined the LoRA framework, including VERA (Kopiczko et al., 2023), which reduces memory usage by sharing the $A$ and $B$ matrices across layers, and rsLoRA (Kalajdzievski, 2023), which scales the learning rate of LoRA $A$ and $B$ adapters by a factor of $\alpha/\sqrt{r}$ instead of original $\alpha/r$ (Hu et al., 2021), where $\alpha$ is a hyperparameter. These developments highlight the growing interest in flexible and efficient adaptation techniques, culminating in the emergence of dynamic LoRA methods. A notable recent approach is DoRA (Mao et al., 2024), which takes a different direction by decoupling the direction and magnitude of rank-one updates. Specifically, DoRA rewrites the LoRA increment as $\Delta W = \alpha \cdot ba^T$, where $b$ and $a$ are unit vectors and $\alpha$ is a learned scalar. The directions are trained during an initial phase and then frozen, which reduces the number of trainable parameters and improves stability, particularly in low-resource scenarios.

### 2.1.1 DYNAMIC LORA RANK ALLOCATION

This topic is of particular relevance in the context of our work, as we focus on adaptive rank learning, enabling the model to automatically adjust the capacity of each adapter during fine-tuning. It has attracted significant attention in recent research Zhang et al. (2023a;b); Sheng et al. (2025), as exploiting dynamically changing ranks allows models to utilize newly allocated parameters more efficiently.

The SVD-like representation of LoRA adapters proposed in AdaLoRA (Zhang et al., 2023b) enables the construction of an importance score for each adapter. This importance score is used to selectively disable certain learning matrices at each fine-tuning step: if an adapter's importance score does not rank among the top-$k$ across all adapters, the corresponding diagonal matrix of singular values is set to zero. To maintain the orthogonality of the singular vector matrices $P$ and $Q$, the authors introduce a regularization term $R(P, Q) = \|P^T P - I\|_F^2 + \|Q^T Q - I\|_F^2$. During training, all three matrices $P$, $\Lambda$, and $Q$ are updated simultaneously. However, this approach has several notable drawbacks. First, the need to compute the SVD of adapter matrices at each fine-tuning

step introduces significant computational overhead, especially for large-scale models. The regular evaluation of importance scores and the associated regularization further increase the complexity and resource requirements of the method. As a result, AdaLoRA can be substantially slower and less scalable compared to other PEFT frameworks, limiting its practical applicability in scenarios where computational efficiency is critical.

In a similar way, Sheng et al. (2025) proposes a rank-one LoRA series representation of the increment matrix: $\Delta \boldsymbol{W} = \boldsymbol{b}_1 \boldsymbol{a}_1 + \ldots + \boldsymbol{b}_p \boldsymbol{a}_p = \begin{bmatrix} \boldsymbol{B}_{p-1} & \boldsymbol{b}_p \end{bmatrix} \begin{bmatrix} \boldsymbol{A}_{p-1} \\ \boldsymbol{a}_p \end{bmatrix}$, where $p$ denotes the number of training loop iterations. This representation is applied to several layers, and after each outer loop step, a convergence criterion is evaluated: if the relative change between the $(p-1)$th and $p$th steps is sufficiently small, the training of $\Delta W$ is terminated. Notably, this approach suffers from the same limitations as IncreLoRA, as it also relies on changes in parameter norms to guide the adaptation process.

Compared to our work, we do not force ranks of LoRA adapters to increase at each step, we rather allow ranks to be trained, hence responding to the needs of a model. We also inherit the ideas of projection and weights freezing in our research.

## 3 SimplexLoRA Framework

### 3.1 Convex Upper Bound

Consider a loss function $\mathcal{L}(\boldsymbol{W}^1, \ldots, \boldsymbol{W}^n)$ for a machine learning model with $n$ layers. In the context of fine-tuning an LLM using the LoRA algorithm, the optimization problem can be formulated as

$$\min_{\mathcal{A}, \mathcal{B}} \left\{ \mathcal{L}(\mathcal{A}, \mathcal{B}) := \mathcal{L}(\boldsymbol{A}^1 \boldsymbol{B}^1, \ldots, \boldsymbol{A}^n \boldsymbol{B}^n) \right\},$$

where $\mathcal{A}$ and $\mathcal{B}$ denote the sets of all LoRA adapters. To generalize this formulation, we introduce a non-negative scaling factor $\omega_i \geq 0$ for each layer $i$, which can be interpreted as rescaling the adapter matrix $\boldsymbol{A}^i$ by $\omega_i$. We further constrain the vector of scaling factors $\omega := (\omega_1, \ldots, \omega_n)$ to lie on the probability simplex $\Delta_{n-1}$, i.e., $\sum_{i=1}^n \omega_i = 1$. The resulting optimization problem is then

$$\min_{\mathcal{A}, \mathcal{B}} \left\{ \mathcal{L}(\omega_1 \boldsymbol{A}^1 \boldsymbol{B}^1, \ldots, \omega_n \boldsymbol{A}^n \boldsymbol{B}^n) \right\}.$$

Note that this reparameterization does not alter the expressive power of the model, as the scaling factors can be absorbed into the adapter matrices.

The classical assumption in the literature is that the optimized function $\mathcal{L}$ is convex (Khaled et al., 2023) with respect to all parameters $\{\boldsymbol{W}^1, \ldots, \boldsymbol{W}^n\}$. While neural networks are generally non-convex, convex analysis remains highly relevant: empirical studies show that neural networks can exhibit locally convex behavior (Kleinberg et al., 2018; Zhou et al., 2019), and many non-convex optimization results build upon convex foundations (Hinder et al., 2020; Fatkhullin et al., 2022). Thus, the convexity assumption provides both theoretical clarity and practical utility for analyzing fine-tuning algorithms in large language models.

Therefore, if we take $\mathcal{A}^i \cdot \mathcal{B}^i := \{0, \ldots, \boldsymbol{A}^i \boldsymbol{B}^i, \ldots, 0\}$, that is, we fine-tune only the $i$-th layer, then, by the convexity of $\mathcal{L}$ (Jensen's inequality), we obtain:

$$\min_{\mathcal{A}, \mathcal{B}} \left\{ \mathcal{L}(\mathcal{A}, \mathcal{B}) \right\} = \min_{\mathcal{A}, \mathcal{B}} \left\{ \mathcal{L}(\omega_1 \boldsymbol{A}^1 \boldsymbol{B}^1, \ldots, \omega_n \boldsymbol{A}^n \boldsymbol{B}^n) \right\}$$

$$= \min_{\mathcal{A}, \mathcal{B}} \left\{ \mathcal{L}\left( \sum_{i=1}^n \omega_i \cdot \{0, \ldots, \boldsymbol{A}^i \boldsymbol{B}^i, \ldots, 0\} \right) \right\} \leq \min_{\mathcal{A}, \mathcal{B}} \left\{ \sum_{i=1}^n \omega_i \mathcal{L}\left( \mathcal{A}^i \cdot \mathcal{B}^i \right) \right\}$$

$$= \sum_{i=1}^n \omega_i \min_{\boldsymbol{A}^i, \boldsymbol{B}^i} \left\{ \mathcal{L}(0, \ldots, \boldsymbol{A}^i \boldsymbol{B}^i, \ldots, 0) \right\}.$$

Thus, the original fine-tuning problem can be upper-bounded by a weighted sum of $n$ other fine-tuning problems, where only the $i$-th layer is trainable and all other layers are frozen. If we then

perform an alternating minimization procedure over $\omega \in \Delta_{n-1}$, the $i$-th element of the resulting weight vector $\omega$ will be greater than the $j$-th element if and only if

$$\min_{\mathcal{A}^i, \mathcal{B}^i} \mathcal{L}(\mathcal{A}^i \cdot \mathcal{B}^i) \leq \min_{\mathcal{A}^j, \mathcal{B}^j} \mathcal{L}(\mathcal{A}^j \cdot \mathcal{B}^j)$$

since otherwise, swapping their positions would yield a smaller value. Therefore, after the alternating optimization procedure $\min_{\omega \in \Delta_{n-1}}$, the layers that are better suited for fine-tuning (i.e., those with a lower minimum) will receive higher scores. This information can then be used to increase the rank of well-adapted LoRA layers and decrease it for "weaker" ones. This is the main idea behind the SimplexLoRA algorithm.

Note that to prevent $\omega$ from degenerating into a one-hot vector, which is a common issue in simplex-constrained optimization where a single component dominates, one can apply regularization. In our implementation, we use a standard Euclidean regularization term to encourage $\omega$ to remain closer to a uniform distribution across all components.

## 3.2 SimplexLoRA Framework

Motivated by these considerations, we introduce a modified algorithm that extends the conventional LoRA framework by adaptively scaling the influence of individual adapters. This modification prioritizes the expansion of critical LoRA adapters while compressing less significant ones, thereby enhancing parameter efficiency.

For the existing LoRA method, we introduce a set of trainable parameters that serve as additional scaling factors for the adapters. This modification can be reformulated as follows:

$$\boldsymbol{W}_0^i + \boldsymbol{A}^i \boldsymbol{B}^i \rightarrow \boldsymbol{W}_0^i + \omega_i \boldsymbol{A}^i \boldsymbol{B}^i,$$

where $\omega_i \geq 0$ represents the learnable importance coefficient assigned to the $i$-th adapter. The importance weights $\omega = (\omega_1, \ldots, \omega_n)$ are constrained to the scaled simplex $n \cdot \Delta_{n-1}$, satisfying $\sum_{i=1}^n \omega_i = n$. This $n$-fold scaling preserves the original magnitude of LoRA adapters; otherwise, the starting matrix weights would be reduced and the starting point would be shifted. Notably, if we initialize $\omega_0 = (1, 1, \ldots, 1) =: \mathbf{1}$, we recover the classical initialization used in LoRA (Hu et al., 2021). This geometric constraint ensures non-negativity and a fixed total mass while preserving the scaling freedom of individual adapters.

The alternating optimization problem in SimplexLoRA is:

$$\min_{\omega \in n \cdot \Delta_{n-1}} \min_{\mathcal{A}, \mathcal{B}} \mathcal{L}(\omega_1 \boldsymbol{A}^1 \boldsymbol{B}^1, \ldots, \omega_n \boldsymbol{A}^n \boldsymbol{B}^n) + \lambda \|\omega - \mathbf{1}\|_2^2, \tag{1}$$

where $\lambda \|\omega - \mathbf{1}\|_2^2$ is the regularization term, and $\lambda \geq 0$ is often referred to as the weight decay.

We solve equation 1 with classical stochastic gradient methods, since all parameters $\omega, \mathcal{A}, \mathcal{B}$ are trainable. Furthermore, note that the dimensionality of the additional vector $\omega$ is $n$, which is significantly smaller than that of each matrix in $\mathcal{A}$ and $\mathcal{B}$. Consequently, the alternating optimization introduces a minimal fraction of trainable parameters. Moreover, the rank learning process is activated only during a small fraction of the total training phase. Once completed, the weights become fixed, introducing no additional computational overhead thereafter. Let us detail the training process of the SimplexLoRA method.

## 3.3 Training Process

The SimplexLoRA algorithm employs an alternating optimization scheme consisting of four distinct phases, which are cyclically repeated for $K$ iterations.

**Phase 1: Warm-up.** For $T$ optimization steps, the model undergoes standard LoRA fine-tuning. Moreover, the weights $\omega$ near the LoRA layers are trained in parallel with all other parameters.

**Phase 2: Simplex projection.** The coefficient vector $\omega$ is projected onto the $n \cdot \Delta_{n-1}$ via Euclidean projection (Blondel et al., 2014), solving the constrained optimization problem:

$$\min_{\omega \in \mathbb{R}^n} \|\omega - \hat{\omega}\|_2^2 \text{ subject to } \omega \geq 0 \text{ and } \sum_{i=1}^n \omega_i = n.$$

This formulation ensures that $\omega$ remains a valid probability distribution (scaled by $n$) while minimizing its Euclidean distance from the unconstrained estimate $\hat{\omega}$. We employ a bisection method, which provides a conceptually simple and efficient solution with worst-case complexity of $\mathcal{O}(n)$. The algorithm iteratively halves the search interval, selecting the subinterval that contains the solution.

While softmax and weighted softmax transformations are commonly used for projecting onto the simplex, we observed severe performance degradation during training when applying these methods. Their behavior is highly sensitive to temperature scaling: high temperatures tend to produce nearly one-hot distributions, whereas low temperatures result in almost uniform outputs. Such degeneracies are particularly problematic in our setting, where the goal is to efficiently allocate the rank budget across multiple adapters without inducing excessive sparsity or uniformity. For these reasons, we utilize the standard Euclidean regularization in the optimization problem formulation equation 1, rather than alternatives such as KL divergence.

**Phase 3: Rank update.** Adapter ranks are dynamically resized while preserving the total parameter budget:

$$r_i^{new} = \lfloor r_0 \cdot \omega_i \rfloor,$$

where $r_0$ denotes the initial rank equal for all adapters (standard LoRA hyperparameter). The floor operation provides integer ranks and does not increase the number of trainable parameters.

**Phase 4: Rank scaling.** Parameter transformations based on rank differences, defined as $\Delta r_i = r_i^{\text{new}} - r_i^{\text{old}}$, are applied to the adapter matrices $\boldsymbol{A}^i$ and $\boldsymbol{B}^i$. When $\Delta r_i > 0$, the rank of the $i$-th adapter is expanded, whereas for $\Delta r_i < 0$, a compression of the $i$-th adapter is performed. A detailed description of the algorithms is provided below.

**Expansion ($\Delta r_i > 0$).** Consider LoRA adapter $\boldsymbol{A}_{old} \cdot \boldsymbol{B}_{old} \in \mathbb{R}^{m \times d}$ with $\boldsymbol{A}_{old} \in \mathbb{R}^{m \times r_{old}}$, $\boldsymbol{B}_{old} \in \mathbb{R}^{r_{old} \times d}$, we wish to extend this old adapter to the rank $r > r_{old}$ and continue learning.

Consider QR factorization:

$$\boldsymbol{A}_{old} = \boldsymbol{Q}_A \boldsymbol{R}_A,$$

where $\boldsymbol{Q}_A \in \mathbb{R}^{m \times r_{old}}$, $\boldsymbol{R}_A \in \mathbb{R}^{r_{old} \times r_{old}}$. It takes $\mathcal{O}(mr_{old}^2)$ to compute QR.

The resulting matrices $\boldsymbol{A}$ and $\boldsymbol{B}$ take the following form:

$$\boldsymbol{A} = [\boldsymbol{Q}_A \ \ (\boldsymbol{I} - \boldsymbol{Q}_A \boldsymbol{Q}_A^T) \boldsymbol{N}] = [\boldsymbol{Q}_A \ \ \tilde{\boldsymbol{N}}_{m \times (r - r_{old})}],$$

$$\boldsymbol{B} = \begin{bmatrix} \boldsymbol{R}_A \boldsymbol{B}_{old} \\ \boldsymbol{O}_{(r - r_{old}) \times d} \end{bmatrix},$$

where random matrix $\boldsymbol{N} \in \mathbb{R}^{m \times (r - r_{old})} \sim \mathcal{N}(0, 1)$.

The matrix product remains invariant under the specified transformation:

$$\boldsymbol{A}\boldsymbol{B} = \boldsymbol{Q}_A \boldsymbol{R}_A \boldsymbol{B}_{old} + \tilde{\boldsymbol{N}} \cdot \boldsymbol{O} = \boldsymbol{A}_{old} \boldsymbol{B}_{old}.$$

**Compression ($\Delta r_i < 0$).** Now we propose methods to reduce the size of the LoRA adapter. Let $\boldsymbol{A}_{old} \in \mathbb{R}^{m \times r_{old}}$, $\boldsymbol{B}_{old} \in \mathbb{R}^{r_{old} \times d}$. We formulate the optimization problem as follows:

$$\min_{\boldsymbol{A}, \boldsymbol{B}} \|\boldsymbol{A}_{old} \boldsymbol{B}_{old} - \boldsymbol{A}\boldsymbol{B}\|_F,$$

where $\boldsymbol{A} \in \mathbb{R}^{m \times r_{new}}$ and $\boldsymbol{B} \in \mathbb{R}^{r_{new} \times d}$. The solution to this problem is the truncated SVD of $\boldsymbol{A}_{old} \boldsymbol{B}_{old}$, which should be computed efficiently.

Let $\boldsymbol{Q}_A \in \mathbb{R}^{m \times r_{old}}$, $\boldsymbol{Q}_B \in \mathbb{R}^{d \times r_{old}}$, and $\boldsymbol{R}_A, \boldsymbol{R}_B \in \mathbb{R}^{r_{old} \times r_{old}}$ be the QR decomposition factors of $\boldsymbol{A}_{old}$ and $\boldsymbol{B}_{old}^{\top}$, respectively. Then we compute:

$$\boldsymbol{U}\Sigma\boldsymbol{V}^T = \text{SVD}(\boldsymbol{R}_A \boldsymbol{R}_B^T),$$

where $U, V \in \mathbb{R}^{r_{old} \times r_{old}}$, and $\Sigma \in \mathbb{R}^{r_{old} \times r_{old}}$. We truncate these matrices to obtain the compressed representation:

$$\begin{cases} U \in \mathbb{R}^{r_{old} \times r_{old}} \\ V \in \mathbb{R}^{r_{old} \times r_{old}} \\ \Sigma \in \mathbb{R}^{r_{old} \times r_{old}} \end{cases} \longrightarrow \begin{cases} U_r \in \mathbb{R}^{r_{old} \times r} \\ V_r \in \mathbb{R}^{r_{old} \times r} \\ \Sigma_r \in \mathbb{R}^{r \times r} \end{cases}$$

The final compressed matrices are then constructed as

$$A = Q_A U_r \in \mathbb{R}^{m \times r},$$
$$B = \Sigma_r V_r^\top Q_B^\top \in \mathbb{R}^{r \times d}.$$

Thus, the operation total time will not exceed $\mathcal{O}(mr_{old}^2 + nr_{old}^2 + r_{old}^3)$, which is negligible since $r_{old} \ll m, d$.

**Repeat phases 1-4 K times.**

**Phase 5: Standard LoRA training with new ranks.** After several repetitions of the previous steps, we perform a final update that involves adding the trained adapters to the original dense layer:

$$W_0^i := W_0^i + \omega_i A_{new}^i B_{new}^i.$$

This is followed by freezing the parameter $\omega$ and reinitializing the adapter weights:

$$\omega \equiv 1, \ A^i \sim \mathcal{N}(0, 1), \ B^i = 0.$$

Thus, we conclude that the ranks have been successfully trained (capturing the principal trends) and the core SimplexLoRA algorithm is considered complete. Following these operations, standard training resumes, during which the $\omega$ weights remain fixed. In future applications, any algorithm utilizing LoRAs can be employed. We provide a schematic illustration of the SimplexLoRA training in Figure 2.

**Note.** We ensure the total number of rank-determining steps, $K \cdot T$, remains sufficiently small. This prevents rank collapse, and allows the main training process to focus primarily on optimizing the LoRA adapters.

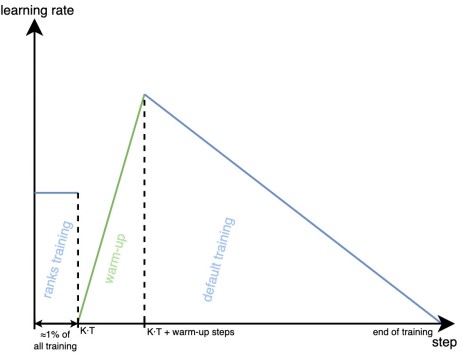

Figure 2: Illustration of the SimplexLoRA training process.

### 3.3.1 SCHEDULER

To ensure that the model resumes training after freezing the weights, we employ a linear scheduler with two training phases (see Figure 3). The first phase includes rank training with reduced learning rate. Once rank training is complete, the learning rate is reset to zero, and the warm-up begins to train the resulting LoRA layers. Note, however, that the learning rate for $\omega$ is fixed during training, the scheduler exclusively adjusts the LoRA weights. As a result, the subsequent fine-tuning phase is functionally identical to the standard LoRA training process.

Figure 3: Scheduler for LoRA parameters with two phases.

### 3.4 MOTIVATING EXAMPLE OF RANK SCALING

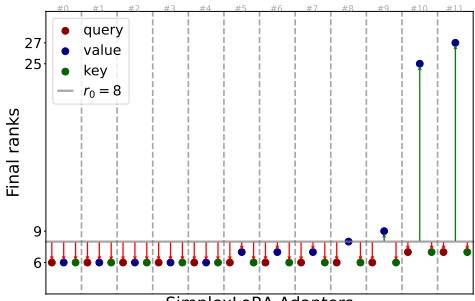

Figure 4 illustrates a representative example of the final rank distribution obtained from fine-tuning a DeBERTaV3$_{\text{base}}$ model (He et al., 2021) on the MRPC dataset from GLUE benchmark (Wang, 2018) with the proposed SimplexLoRA framework, where only the self-attention layers are adapted. The gray line indicates initial rank values, while green and red arrows represent LoRA rank modifications (increases and decreases, respectively).

Figure 4: Illustration of the rank distribution of SimplexLoRA.

The learned weight distribution highlights a clear pattern: the value projections in self-attention layers consistently emerge as the most critical components, as evidenced by their substantially higher allocated ranks. Furthermore, we emphasize a key characteristic of our algorithm: rather than eliminating ranks of adapters with low weights, it reduces their magnitude. This behavior helps to mitigate the problem of underfitting.

## 4 EXPERIMENTS

We implement SimplexLoRA for fine-tuning DeBERTaV3$_{\text{base}}$ (He et al., 2021) and Llama 3.1 8B (AI@Meta, 2024). We evaluate effectiveness of the proposed algorithm on natural language understanding tasks from GLUE benchmark (Wang, 2018). The dataset was chosen based on its established prevalence and broad adoption within the research community.

### 4.1 EXPERIMENTAL SETTINGS

Core algorithm of the proposed method is implemented with PyTorch (Paszke et al., 2019), Huggingface library. All the experiments are conducted on NVIDIA GeForce RTX 2080 Ti GPUs. Detailed hyperparameters values can be found in Appendix A. Let us just note that following LoRA we use a scaling factor $\alpha/r$ (Hu et al., 2021) for adapters ($\alpha = 32$ and $r$ updates correspondingly to $r_i^{new}$). Due to the specific logic required for updating adapter ranks, we implemented the modified version of the AdamW optimization algorithm (Loshchilov, 2017), which included steps with an increased learning rate for weights $\omega$, projection of weights onto the simplex, dynamic rank updates and custom scheduler.

### 4.2 BASELINES

We compare SimplexLoRA with the following methods:

• *Full fine-tuning* is the most common baseline to compare with. All pre-trained weights undergo gradients updates.

• *PEFT* methods from PEFT library include several approaches. All the methods inherit state-of-the-art in reparametrisation-based parameter-efficient fine-tuning approach LoRA. The number of trainable parameters is controlled by the rank $r$.

Our comparison is conducted in two stages. First, we evaluate a broad range of methods in their default configurations (i.e., without hyperparameter optimization) using a limited number of training epochs (see Table 1). Subsequently, we select the top-performing models and perform a more comprehensive comparison, including hyperparameter tuning and an increased number of training epochs (see Table 2 and 3).

## 4.3 EXPERIMENTAL RESULTS

### 4.3.1 "OUT-OF-THE-BOX" COMPARISON

In Table 1 we present evaluation results on the DeBERTa model for the all mentioned methods. This table summarizes experimental runs without parameter tuning. Within these experiments adapters are inserted into self-attention layers only; query, key, and value projections are modified. We use publicly available implementations to run all the baselines.

As demonstrated by our experiments, the most effective methods among those evaluated are full fine-tuning, standard LoRA, and SimplexLoRA. The remaining approaches require stratified parameter tuning to achieve comparable or superior performance.

Table 1: Comparative performance of "out-of-the-box" algorithms on DeBERTa.

| Method | # Params | MNLI Acc | SST-2 Acc | CoLA Mcc | QQP Acc/F1 | QNLI Acc | RTE Acc | MRPC Acc/F1 | ALL Avg |
|---|---|---|---|---|---|---|---|---|---|
| Full Fine-Tuning | 184M (100%) | 0.8910 | 0.9541 | 0.6806 | 0.8962/0.8644 | 0.9383 | 0.8376 | 0.8848/0.9165 | 0.8689 |
| LoRA$_{r=8}$ (Hu et al., 2021) | 442K (0.24%) | 0.8797 | 0.9450 | 0.6913 | 0.8802/0.8437 | 0.9301 | 0.8448 | 0.8897/0.9220 | 0.8655 |
| LoHA$_{r=8}$ (Hyeon-Woo et al., 2021) | 884K (0.48%) | 0.8560 | 0.9392 | 0.6295 | 0.8674/0.8269 | 0.9085 | 0.8051 | 0.8628/0.9007 | 0.8382 |
| Krona$_{r=32}$ (Edalati et al., 2022) | 57.6K (0.03%) | 0.7587 | 0.9289 | 0.5907 | 0.8275/0.7816 | 0.8605 | 0.7509 | 0.7500/0.8416 | 0.7843 |
| AdaLoRA$_{r=8}$ (Zhang et al., 2023b) | 664K (0.36%) | 0.8390 | 0.9392 | 0.6222 | 0.8534/0.8083 | 0.8999 | 0.7834 | 0.7059/0.8225 | 0.8113 |
| VERA$_{r=1024}$ (Kopiczko et al., 2023) | 1.64M (0.88%) | 0.8355 | 0.9404 | 0.6252 | 0.8592/0.8203 | 0.8975 | 0.7581 | 0.8529/0.8962 | 0.8244 |
| rsLoRA$_{r=8}$ (Kalajdzievski, 2023) | 442K (0.24%) | 0.8814 | 0.9564 | 0.6414 | 0.8853/0.8506 | 0.9325 | 0.8267 | 0.8750/0.9122 | 0.8571 |
| DoRA$_{r=8}$ (Liu et al., 2024) | 442K (0.24%) | 0.8811 | 0.9473 | 0.6748 | 0.8809/0.8451 | 0.9295 | 0.7978 | 0.8799/0.9139 | 0.8558 |
| SimplexLoRA$_{r=8}$ | 442K (0.24%) | 0.8984 | 0.9587 | 0.6572 | 0.9124/0.8844 | 0.9362 | 0.8339 | 0.8987/0.9289 | **0.8709** |

Given these findings, we select LoRA as the most efficient baseline for comparison with SimplexLoRA, as it does not necessitate intricate hyperparameter tuning. In practice, such fine-grained adjustments are often avoided due to their high sensitivity, excessive computational costs, or impractical time requirements for exhaustive search.

### 4.3.2 COMPARISON TO LoRA

In Table 2 we report evaluations on the top-performing fine-tune algorithm LoRA with four different ranks on DeBERTa. During these evaluations adapters are inserted into all linear layers. It is worth noting that in most experiments, SimplexLoRA outperforms the standard algorithm while maintaining the same computational cost in terms of tuning and training time.

To validate the generalizability of our approach across different architectures, we additionally evaluated SimplexLoRA on the Llama. We selected rank 2 as a starting point for both algorithms. As can be seen from Table 3, our approach does not fall short of LoRA and even surpasses it in some cases. The observations presented in Tables 1, 2, and 3 indicate that our method yields results comparable to the baseline approaches, while outperforming the majority of them on most downstream tasks. All improvements are proved to be statistically significant with corresponding statistical tests.

Although our algorithm does not directly propose methods for reducing the number of trainable parameters, it can be observed that lowering the initial ranks enables such reduction while preserving accuracy (e.g., ranks 1 and 2 from Table 2).

Table 2: Comparative results of top-performing algorithms: LoRA and SimplexLoRA on DeBERTa.

| Rank | Method | # Params | MNLI Acc | SST-2 Acc | CoLA Mcc | QQP Acc/F1 | QNLI Acc | RTE Acc | MRPC Acc/F1 | ALL Avg |
|---|---|---|---|---|---|---|---|---|---|---|
| $r = 1$ | LoRA | 0.07% | 0.8912 | 0.9488 | 0.6558 | 0.8905/0.8544 | 0.9250 | 0.7943 | 0.8905/0.9213 | 0.8562 |
| | SimplexLoRA | | 0.8913 | 0.9576 | 0.6775 | 0.9018/0.8709 | 0.9317 | 0.8159 | 0.9093/0.9359 | **0.8690** |
| $r = 2$ | LoRA | 0.13% | 0.9004 | 0.9413 | 0.6600 | 0.9077/0.8631 | 0.9369 | 0.8068 | 0.9003/0.9193 | 0.8629 |
| | SimplexLoRA | | 0.8946 | 0.9608 | 0.6658 | 0.9178/0.8842 | 0.9338 | 0.8252 | 0.9103/0.9314 | **0.8717** |
| $r = 4$ | LoRA | 0.26% | 0.8780 | 0.9562 | 0.6809 | 0.9153/0.8866 | 0.9582 | 0.7816 | 0.9207/0.9471 | 0.8700 |
| | SimplexLoRA | | 0.9002 | 0.9610 | 0.6807 | 0.9244/0.8903 | 0.9358 | 0.8259 | 0.9044/0.9322 | **0.8756** |
| $r = 8$ | LoRA | 0.51% | 0.9030 | 0.9641 | 0.6965 | 0.9092/0.8723 | 0.9449 | 0.8537 | 0.9064/0.9127 | 0.8804 |
| | SimplexLoRA | | 0.9047 | 0.9619 | 0.6978 | 0.9129/0.8901 | 0.9447 | 0.8305 | 0.9267/0.9475 | **0.8826** |

Table 3: Comparative results of top-performing algorithms: LoRA and SimplexLoRA on Llama.

| Method | # Params | MNLI Acc | SST-2 Acc | CoLA Mcc | QQP Acc/F1 | QNLI Acc | RTE Acc | MRPC Acc/F1 | ALL Avg |
|--------|----------|------|-------|------|----------|------|-----|----------|------|
| LoRA | 0.13% | 0.9029 | 0.9712 | 0.6641 | 0.9227/0.8994 | 0.9406 | 0.8812 | 0.9158/0.9306 | 0.8849 |
| SimplexLoRA | | 0.9043 | 0.9802 | 0.6715 | 0.9301/0.9072 | 0.9411 | 0.9109 | 0.9307/0.9517 | **0.8954** |

## 4.4 ABLATION STUDY

As a supplementary study, we conducted a limited hyperparameter grid search on the MNLI dataset to evaluate the robustness of our algorithm (see Figure 5). In this experiment, both SimplexLoRA and standard LoRA were trained for 2048 steps. The figure cells contain the final difference between SimplexLoRA and LoRA accuracy on validation.

The results demonstrate that our algorithm maintains consistent performance within a constrained optimization budget (i.e., when $K \cdot T < 100$, which is less than 5% of the total training). However, excessively large hyperparameter values should be avoided, as they may lead to rapid rank overfitting. Specifically, when using small values such as $K = 3$ and $T = 5$, the ranks do not have sufficient time to converge and adapt meaningfully, leading to underfitting. On the other hand, increasing the number of projections to $K = 10$ or the optimizer steps to

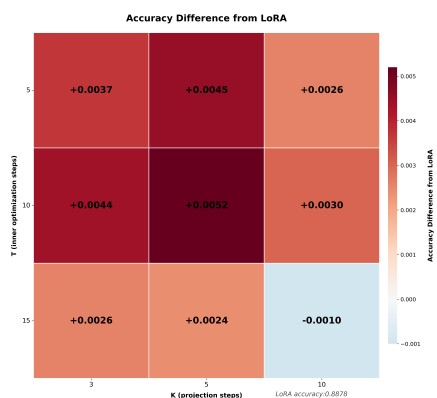

Figure 5: The influence of hyperparameters on SimplexLoRA performance.

$T = 15$ often results in overfitting. These observations highlight the importance of keeping the total number of steps $K \cdot T$ relatively low. In all experiments, we used $K = 5$ and $T = 10$, which consistently delivered strong performance.

## 5 LIMITATIONS

The limitations of the proposed strategy are as the following. First, it introduces three additional hyperparameters: the number of iterations between simplex projection steps, the total number of such projections, and the weight decay coefficient used for regularization. Second, the lack of experiments on larger-scale models limits the generalizability of our findings. Finally, a fundamental ambiguity remains during training: a small gradient norm in a layer may indicate either that the weights are well-optimized or that the gradients have simply vanished by the time they reach that layer.

## 6 CONCLUSION

This work presents SimplexLoRA, a parameter-efficient fine-tuning method that adaptively learns the ranks of LoRA adapters during training. By dynamically reallocating the rank budget, our approach identifies and prioritizes the most important layers, while less critical adapters receive reduced capacity. The rank allocation is performed in a brief initial phase, after which the learned ranks are fixed and standard LoRA fine-tuning proceeds, ensuring compatibility with existing LoRA-based techniques. Our experiments on the GLUE benchmark demonstrate that SimplexLoRA consistently matches or surpasses state-of-the-art baselines, while the rank optimization phase accounts for less than 5% of the total training time. This highlights the practical applicability and efficiency of our method. Finally, we provide efficient matrix compression and expansion strategies to support dynamic rank adjustment without increasing the overall parameter budget. Overall, SimplexLoRA offers a robust and scalable solution for resource-constrained fine-tuning scenarios.

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

## A    EXPERIMENTAL DETAILS

To ensure reproducibility of the experiments, we provide additional details regarding the number of training epochs and batch size that we used in Table 2 (see Table 4). The experiments from Table 1 were obtained using fixed 512 iterations with the same batch size, while those in Table 3 used 1024 iterations and a batch size of 16.

We also provide additional information about hyperparameters that we used. AdamW parameters includes $\beta_1 = 0.9$, $\beta_2 = 0.999$, weight decay for parameters is set to zero. Learning rate was tuned using grid: $\{3 \cdot 10^{-5}, 5 \cdot 10^{-5}, 8 \cdot 10^{-5}, 1 \cdot 10^{-4}, 3 \cdot 10^{-4}, 5 \cdot 10^{-4}, 8 \cdot 10^{-4}, 1 \cdot 10^{-3}\}$. Learning rate for $\omega$ was increased and set to $5 \cdot 10^{0}$. For the comparison to LoRA experiments and the ablation study we tuned warmup steps using grid: $\{0, 20, 30, 50, 70, 100, 200\}$, regularization term weight decay using grid: $\{1 \cdot 10^{0}, 1 \cdot 10^{-1}, 1 \cdot 10^{-2}, 1 \cdot 10^{-3}, 1 \cdot 10^{-4}, 1 \cdot 10^{-5}, 1 \cdot 10^{-6}\}$.

Additionally, we used 6 gradient accumulation steps for DeBERTa and 1 accumulation step for Llama across all methods. For LoRA and PEFT, we employed a linear scheduler, while SimplexLoRA used a scheduler with 2 training phases. We set dropout to $p = 0.05$.

Table 4: Hyper-parameter setup for DeBERTa model on GLUE benchmark.

| Task | # epochs | batch size |
|------|----------|------------|
| MNLI | 7 | 32 |
| SST-2 | 10 | 32 |
| CoLA | 15 | 64 |
| QQP | 5 | 32 |
| QNLI | 5 | 32 |
| RTE | 50 | 64 |
| MRPC | 30 | 64 |

## B    THE USE OF LARGE LANGUAGE MODELS (LLMS)

During the preparation of this work, the authors used a LLM to assist with several tasks. Specifically, the LLM was employed for code generation to create data visualization scripts, debugging existing code, and for proofreading the paper to improve grammatical accuracy and stylistic consistency. After using this tool, the authors reviewed and edited the content as needed for the final publication.

