# OpenReview forum: "SimplexLoRA: Dynamic Rank Updating via Alternating Minimization with Simplex Projection"
_ICLR.cc/2026/Conference — ICLR 2026 Conference Withdrawn Submission_

### Official Review · Reviewer_Qxta · 2025-10-18

**Soundness:** 2
**Presentation:** 2
**Contribution:** 2
**Rating:** 4
**Confidence:** 5

**Summary:**

SimplexLoRA is a variant of LoRA that dynamically adjusts each layer’s adapter rank during fine-tuning. The method works by first training the network with a learnable scalar before each LoRA adapter AB, then normalizing these scalars via simplex projection to form a probability distribution representing the relative importance of layers, and finally adjusting each layer’s rank proportionally to its normalized scalar.

**Strengths:**

* Interesting and simple method
* The method shouldn’t increase training time too much
* Table 3, the improvement is good: 89.54 vs 88.49, but since we don’t have in the Lora/AdaLora paper a score for Llama, it would have been far more convincing if we had: (a) stdev across three consecutive seed numbers (b) Comparison to at least AdaLora as well (c) code for reproducibility to verify the results

**Weaknesses:**

* Table 1: The reader must understand more about the hyperparams. You wrote “without parameter tuning”. That means that you took the best hyperparameters from the original baseline paper? What are they? Or that you decided on the same hyperparams for all benchmarks, and if so how and why?


* Table 1: You wrote that AdaLora received 81.13, but in the AdaLora paper they say they received 89.31, which is a huge difference.
* Table 2: The main point in the proposed method is the different rank per layer. I believe the method should not only be compared to LoRA, but also to AdaLoRA, PriLoRA, etc.
* Table 2: It is common to also add stdev when possible, otherwise a single seed can be cherry picked in order to win. Also, no code was supplied for reproducibility and double checking of the results. You wrote that AdaLora got 88.04 (less than your method 88.26), while in AdaLora paper they write 88.34 (higher than your method 88.26). That’s the reason you need stdev.



* In table 1 and others, no need to have all these leading zeros before the numbers. It makes reading more difficult. (See LoRa, AdaLora and others )

**Questions:**

* Table 1: What about the STS-B benchmark which is part of GLUE? (See LoRA, AdaLoRA, PRILoRA and others). It's okay that you don't do the full GLUE set, but at least say something about it, and don't let the reader guess.

---

### Official Review · Reviewer_mYfw · 2025-10-27

**Soundness:** 2
**Presentation:** 2
**Contribution:** 2
**Rating:** 2
**Confidence:** 3

**Summary:**

The paper studies the question of how to find the optimal ranks for adapters used in LoRA. This paper frames this as optimizing a weighted combination of the adapters, where the weights are trainable and live in the simplex. The ranks are dynamically assigned during the training, proportional to the weights. The paper also specifies how to extend and contract the matrices $A,B$ when the rank increases or decreases. In experiment, the method is shown to have performance gains over existing approaches.

**Strengths:**

- The question of the rank assignment in LoRA is a valid and interesting question and it is a difficult problem. The proposed method has the advantage of being simple compared with some of the existing approaches.
- The experiment show pretty good results.

**Weaknesses:**

- The derivation in Line 153 - 159 doesn't really reflect the nature of the problem. The implicit assumption here (which should be mentioned explicitly ) is that the ranks of $A_i, B_i$ are fixed. Also this upperbound suggests that the adapters are tuned separately, which is not the case in the paper.
- The paper misses some baselines.
  - Zhang, Qiang, Somayajula, Xie: AutoLoRA: Automatically Tuning Matrix Ranks in Low-Rank Adaptation Based on Meta Learning
  - Zhou, Wan, Vulic, Korhonen: AUTOPEFT: Automatic Configuration Search for Parameter-Efficient Fine-Tuning
- The time overhead to perform QR decomposition is not studied.
- Lacking experiments: Apart from the lack of experiments on bigger model, there are other benchmarks, such as the commonsense dataset, etc, that standard fine-tuning methods usually use that this paper misses out on.

**Questions:**

- Regarding the rank expansion: Why don't we simply use $A = [A^{old} | N]$ and $B^\top = [B^{old, \top} | 0]$ but needs to go by the QR decomposition?

---

### Official Review · Reviewer_Jacn · 2025-11-03

**Soundness:** 1
**Presentation:** 3
**Contribution:** 2
**Rating:** 2
**Confidence:** 4

**Summary:**

To adaptively adjust adapter ranks based on their importance during LoRA fine-tuning, this paper proposes SimplexLoRA, which learns a set of per‑adapter weights $\omega$ constrained to being non-negative and sum to the number of adapters (scaled simplex). During a short “rank‑learning” phase, $\omega$ is optimized with stochastic gradient methods and then projected onto the simplex via Euclidean projection, and the rank is scaled by $\omega$. To change ranks on the fly, the method provides rank‑expansion (QR + random orthogonal completion) and rank‑compression (QR + truncated SVD on $R_AR_B^\top$) operators. On GLUE with DeBERTa‑V3‑base and Llama‑3.1‑8B, SimplexLoRA matches or surpasses LoRA at similar parameter budgets, with visible gains at very low ranks (e.g., r=1,2).

**Strengths:**

* The proposed method is simple and easy to use in many sense: Only one scaling factor is learned per adapter; the rank-learning phase is very short and the overhead is negligible; After the rank-learning phase it reverts to standard LoRA training, and thus should combine easily with other LoRA variants and techniques.
* This method comes with an efficient operators to change adapter rank on the fly. The expansion operator increases rank while preserving the product $\boldsymbol{AB}$ exactly, and the compression operator decreases rank while giving the best Frobenius‑norm approximation to the product $\boldsymbol{AB}$.

**Weaknesses:**

* The theoretical assumption is too strong. The SimplexLoRA framework adopts a convex setting, while neural networks are generally non-convex. While convexity assumption can be useful, in the setting of Sec. 3.1, it will lead to trivial solutions where only one adapter should ever exist, which does not make sense and indicates the theory is too crude to solve this problem. Consequently, a regularization term is needed in eq. (1), which does not fit into the theory.
* The assumption is that $\omega$ reflects the importance of the adapter. However, invariant is achieved by multiplying $\boldsymbol{A}$ or $\boldsymbol{B}$ by a factor and divide $\omega$ by the same factor.
* Experiment in 4.3.1 can be misleading. LoRA is "top-performing" because the default hyperparameters are tuned on LoRA and does not mean the method itself is better and can be a decent baseline.
* Even compared with the plain LoRA, the improvements of SimplexLoRA is limited. In table 2, the improvements diminishes as the rank increases. In practice, one can use a rather large rank without introducing much overhead, and it nullifies the motivation of using the proposed method.

**Questions:**

* Is SimplexLoRA in table 1 tuned, or it uses exactly the same hyperparameters as plain LoRA? Are the three new hyperparameters tuned?
* How about just using the norm of $\boldsymbol{AB}$ (maybe also go through softmax or Euclidian projection) to replace $\omega$ as an importance metric to scale the rank, which should be more straightforward?
* In the proposed algorithm, can the rank of an adapter be 0?

---

### Official Review · Reviewer_Ti1o · 2025-11-05

**Soundness:** 2
**Presentation:** 2
**Contribution:** 2
**Rating:** 4
**Confidence:** 3

**Summary:**

This paper proposes attaching a weight parameter $\omega_i$ to each LoRA adapter and train these weights together with the adapter under the constraint that they lie on the scale simplex $\sum \omega_i = n$. The weights are used to update the rank for each LoRA adapter where the $i$-th adapter has rank $\omega_i r_0$ where $r_0$ is the initial rank for all adapter.

With a convexity assumption the authors show that a high $\omega_i$ signals that the layer is better suited for finetuning because it can achieve a low loss on its own given all other layers remaining unchanged.

Experiments on the GLUE benchmark with DeBERTaV3 and Llama 3 models show that SimplexLoRA matches or surpasses the performance of standard LoRA and other adaptive methods, often while significantly reducing the total number of trainable parameters.

**Strengths:**

- Motivated by that if we only do LoRA on a single layer, those layers that achieve a low loss should be allocated higher rank, the authors come up with a new loss formulation to determine which layer needs higher rank.
- The authors ensure the total numbers of rank determining steps K * T small enough so the training process still mainly focuses on the LoRA adapters

**Weaknesses:**

- IIUC, the authors assume that if a layer and achieves a low loss on its own (given all other layers remaining the same without LoRA), we should allocate a higher rank for it. Are there any insights why it is? It could be that the layer is doing its job correctly without being finetuned and can be assign a low rank. Since this is the main motivation for the method I think it isn't strongly backed up.
- Table 1 doesn't seem fair since most methods other than LoRA doesn't have their optimal setup, resulting in all of them having lower results than LoRA, while SimplexLoRA introduces some hyper-parameter that may have been tuned. Table 2 only compares SimplexLoRA with LoRA on a few different setups but doesn't include other methods.

**Questions:**

- Can you give more insights about the assumption in the weaknesses?

---

### Note · Authors · 2026-02-03

I have read and agree with the venue's withdrawal policy on behalf of myself and my co-authors.

---

### Meta-Review · Area_Chair_Pdqp · 2026-01-07

**Summary:**

The reviewers identified fundamental weaknesses in both the theoretical framing and the empirical rigor of the submission.

**Theoretical Validity:** A primary concern shared by multiple reviewers was the paper's core assumption: that adapters achieving lower loss in isolation should necessarily be allocated a higher rank. Reviewers noted this intuition relies on convexity-style arguments that do not transfer well to the non-convex nature of neural network training, leaving the method's theoretical justification weak.

**Experiment:** Significant issues were raised regarding the experimental setup. Reviewers noted the presence of missing or inconsistent baselines, which made it challenging to accurately assess the true performance gains. Additionally, the lack of variance reporting and unclear hyperparameter tuning raised concerns about reproducibility.

**Efficiency:** The computational overhead of the proposed rank update operations was noted as "unexamined," with reviewers questioning the practical viability of the method.

The authors did not participate in the rebuttal process. Consequently, no clarifications were provided, and no revisions were made to the manuscript.

**Reviewer Concerns:**

All concerns remain outstanding, as the authors did not participate in the rebuttal process

**Reviewer Scores:**

The reviewers likely would have maintained or lowered their scores.

---

### Decision · Program_Chairs · 2026-01-26

Reject